# Parental Willingness and Associated Factors of Pediatric Vaccination in the Era of COVID-19 Pandemic: A Systematic Review and Meta-Analysis

**DOI:** 10.3390/vaccines10091453

**Published:** 2022-09-02

**Authors:** Zixin Wang, Siyu Chen, Yuan Fang

**Affiliations:** 1Jockey Club School of Public Health and Primary Care, Faculty of Medicine, The Chinese University of Hong Kong, Hong Kong 666888, China; 2Department of Health and Physical Education, The Education University of Hong Kong, Hong Kong 999088, China

**Keywords:** childhood/routine vaccination, seasonal influenza vaccination, human papillomaviruses vaccination, pneumococcal conjugate vaccination, children, parental willingness

## Abstract

A significant decline in pediatric vaccination uptake due to the COVID-19 pandemic has been documented. Little is known about the parental willingness and associated factors of pediatric vaccination during the COVID-19 pandemic. An extensive literature search in the databases of PubMed, Scopus, Web of Science, and EBSCOhost were conducted. A total of 20 eligible studies published from 2020–2022 were included for systematic summary by a thematic analysis, among which 12 studies were included in a meta-analysis conducted with R-4.2.1. The prevalence of parental willingness to childhood/routine vaccination and seasonal influenza vaccination was 58.6% (95%CI 2.8–98.6%) and 47.3% (95%CI 25.3–70.5%). Moreover, there is no sufficient evidence of significant change in parental willingness to childhood/routine vaccination, human papillomavirus vaccination, or pneumococcal conjugate vaccination during the pandemic. However, a significant increase in parental willingness to vaccinate their children against seasonal influenza was found. In addition to the factors of parental vaccination willingness/hesitancy that are well-studied in literature, children/parents’ history of COVID-19 and children’s perceived vulnerability to COVID-19 were associated with parental willingness. Developing synergetic strategies to promote COVID-19 vaccination together with other pediatric vaccination is warranted during the pandemic. This may help to improve and/or catch up the vaccine uptake of children during and/or after the COVID-19 pandemic.

## 1. Introduction

Vaccines have a positive impact on children’s health and development [1]. Based on data reported by the World Health Organization (WHO), pediatric vaccination averts the annual deaths of 2.5 million children younger than 5 years old worldwide [1]. Although the national immunization programs across countries have achieved a great progress in increasing vaccine coverage, parental vaccine hesitancy, and/or refusal still jeopardizes these programs. Factors associated with decision-making in vaccination were well-studied by several conceptual models in order to design evidence-based interventions for immunization promotion. These conceptual models include the Vaccine Hesitancy Determinants Matrix established by the WHO SAGE working group [2], the behavioral–ecological model of vaccination decision making [3], the models of vaccine hesitancy determinants following the Theory of Planned Behavior [4], or the Health Belief Model [5].

The outbreak of the coronavirus disease 2019 (COVID-19) had a great negative influence on pediatric vaccination on a global scale. A recently published study documented a decline in the number of administered doses of diphtheria tetanus toxoid and pertussis (DTP3) and measles-containing vaccine first dose (MCV1) worldwide in June 2020 [6]. The largest decline in DTP3 uptake occurred in April 2022 (33%) [6]. In response to this, the WHO and the UNICEF issued a warning in regard to this dangerous decrease in children’s vaccinations during the COVID-19 pandemic [7]. The WHO regional office attributed such a phenomenon to the interruption of vaccination demand and supply and the shortage of healthcare workers. Moreover, routine vaccination for children may not be the top priority of medical service among healthcare providers during the pandemic [8]. Even when such services are offered, people may be either unable to access them due to COVID-19 control measures (lockdown, travel restriction), or fear of being exposed to people with COVID-19 [8]. Many health workers are also unavailable due to COVID-19 control measures and a lack of protective equipment [8]. However, there is a vague image showing the influence of the COVID-19 pandemic on pediatric vaccination hesitancy among parents, who are usually the decision makers for their children’s vaccination.

To address the knowledge gaps, this study aimed to systematically search and summarize the evidence from literature reporting parental willingness/hesitancy about the vaccination uptake in their children in the time of COVID-19 (in terms of childhood/routine vaccination, seasonal influenza vaccination, human papillomavirus (HPV) vaccination, and pneumococcal conjugate vaccination (PCV)). In addition, we also summarized the factors associated with parental willingness/hesitancy of pediatric vaccination during the COVID-19 pandemic.

## 2. Methods

### 2.1. Search Strategy

This systematic review was registered in PROSPERO (ref#: PROSPERO 2022 CRD42022350081). The academic articles were identified by searching the electronic databases PubMed, Web of Science (including SSCI and A&HI), EBSCOhost (including CINAL with Full Text, ERIC, MEDLINE, APA PsycArticles and APA PsycINFO), and Scopus (including both published papers and preprinted services), which cover the published periods of 1966–2022, 1975–2022, 1981–2022, and 1970–2022 respectively. The Boolean operator was used in the search strategy conducted, with “OR” and/or “AND” used to link search terms, while the asterisk “*” was used as a wildcard symbol appended at the end of the terms to search for variations of those terms. The complete search strategy is described below:(a)“impact” or “effect” or “influence” or “outcome*” or “result*” or “consequence” in topic/title/abstract;(b)“willingness” or “intention” or “acceptance” or “attitude” or “perception” in topic/title/abstract;(c)“COVID-19” or “coronavirus” or “2019-ncov” or “SARS-CoV-2” or “cov-19” in topic/title/abstract;(d)“pandemic” or “epidemic” or “outbreak” in topic/title/abstract;(e)“pediatric” or “child*” or “infant” or “adolescent” or “early childhood” or “routine” in topic/title/abstract;(f)“vaccin*” or “immunizations” in topic/title/abstract;(g)a OR b;(h)c AND d;(i)e AND f;(j)g AND h AND i.

We reviewed articles for relevance through screening titles and abstracts. After removing the duplicates and excluding those who did not meet the inclusion criteria, the remaining 31 articles were read in full to identify if they met the inclusion criteria. In addition, another research paper was included by checking through the reference lists of previous literature [9,10,11,12,13,14]. Finally, there were 20 relevant articles identified.

### 2.2. Inclusion and Exclusion Criteria

The articles included in this review are original studies that were published in peer-reviewed journals in English using quantitative and/or qualitative methodology, and therefore providing a higher quality of evidence than a typical case study or case series, which tends to be more subjective. The eligible studies should report the willingness, intention, attitudes, acceptance, or perception of parents/caregivers toward pediatric/childhood/routine vaccination of their children during the COVID-19 pandemic. The participants in the reviewed studies were parents and/or caregivers of children. An article was excluded if it was (1) not clarified to be designed and conducted during the COVID-19 pandemic; (2) a study for the development or validation of scale/tool; (3) a study for the development of a study protocol; (4) a study that was unclear about whether it included parents/caregivers of children; (5) a study focused on the uptake/administration/coverage of pediatric/childhood/routine vaccination; (6) a study focused on modeling or projection for prediction of vaccination coverage; and/or (7) a study focused on perception/attitude of health professionals or stakeholders.

### 2.3. Quality Assessment and Data Analysis

We used the tool developed by Hawker et al. to assess the quality of the studies. The tool covers nine domains: (1) abstract and title (did they provide a clear description of the study?), (2) introduction and aims (was there a good background and clear statement of the aims of the research?), (3) method and data (is the method appropriate and clearly explained?), (4) sampling (was the sampling strategy appropriate to address the aims?), (5) data analysis (was the description of the data analysis sufficiently rigorous?), (6) ethics and bias (how have ethical issues been addressed, and what has necessary ethical approval gained? Has the relationship between researchers and participants been adequately considered?), (7) results (are these a clear statement of the findings?), (8) transferability and generalizability (are the findings of this study transferable/generalizable to a wider population?), and (9) implications and usefulness (how important are these findings to policy and practice?). These domains are rated 1 = very poor, 2 = poor, 3 = fair, and 4 = good. The detailed assessment criteria for each domain were listed in the footnote of Appendix A [15]. Furthermore, these eligible studies were listed under a structured frame, i.e., study design, samples, and key measurement(s). The results are shown in Appendix A. The extracted information from the listed references was analyzed and summarized using a thematic approach to present an overall scope and compare the targeted information under the key themes. The two authors (Z.W. and Y.F.) independently completed the review process and data extraction. We reached a consensus about the final inclusion, analysis, and summary via discussion.

The meta-analysis was conducted with the metaphor package of R version 4.2.1 (R Core Team, 2022, Vienna, Austria) to combine data, calculated pooled proportions, and 95% confidence intervals (CI). During analysis, a random-effects model was used based on the generalized linear mixed-effects method. The *I*^2^ statistic test was used to measure heterogeneity. *I*^2^ values of <25%, 25 to 75%, and > 75% indicate low, moderate, and high heterogeneity, respectively. Furthermore, visual inspection was employed to assess the asymmetry of funnel plots and the Egger’s test was applied to detect potential publication bias. Sensitivity analysis was conducted by removing one study each time.

## 3. Results

The flowchart describing the search process is presented in Figure 1, which outlined a detailed overview of the search process as well as the reasons for exclusion in this review.

### 3.1. Overview of Included Studies

A total of 20 papers published between 2020–2022 were included in the final analysis and summary, while the full list and the key information of the included studies are shown in Table 1. These studies reported the willingness/attitudes towards pediatric vaccination and associated factors in 12 countries during the COVID-19 pandemic, including the United States (6), China (4), Saudi Arabia (4), Turkey (2), Switzerland (2), Indonesia (1), Mozambique (1), Albania (1), Canada (1), Israel (1), Japan (1), and Spain (1). With the exception of one cross-sectional in-depth interview, quantitative data was collected in other 19 studies via cross-sectional surveys. Most of the studies used a self-developed questionnaire and one study used the Vaccine Hesitancy Scale (VHS) [16]. The samples in the reviewed studies included 22,902 parents or caregivers of children. All participants were ≥20 years old, and 46–100% were mothers. In eleven studies, more than half of parents received tertiary education. Two studies included caregivers of children with medical conditions (i.e., 78 parents of children diagnosed with asthma [17]; and 2422 caregivers of children who received treatment in the emergency room) [18]. Other background information is listed in Table 1.

Moreover, 12 of 20 studies were further involved in meta-analysis. It comprised 17,455 parents and/or caregivers of children, and a random effect model was utilized in analysis. Overall, no publication bias was found according to the Egger’s test as the *p* value ranged from 0.34–0.79 in all analyses (Funnel plots are shown in Appendix A).

### 3.2. Pediatric Vaccination Hesitancy/Willingness and Associated Factors

The prevalence of parental willingness of childhood/routine vaccination, seasonal influenza vaccination, PCV and HPV vaccination, as well as the associated factors in this review, are summarized in Figure 2, Table 1, Appendix A.

#### 3.2.1. Childhood/Routine Vaccination

According to the meta-analysis, the pooled prevalence of hesitancy towards childhood/routine vaccination was 18.5% (95% CI = 9.0–34.3%, four studies, *I*^2^ = 99%; Appendix A), whereas the pooled prevalence of willingness towards childhood/routine vaccination was 58.6% (95% CI = 2.8–98.6%, two studies, *I*^2^ = 97%; Appendix A). In particular, five of them reported hesitancy towards childhood vaccination with huge diversity. A repeated cross-sectional survey showed a fluctuant trend in parental vaccine hesitancy (*p* trend < 0.01), i.e., 7.8% during September–October 2020, 15.1% during February–March 2021, and 5.5% during May–Jun 2021 [16]. Moreover, the prevalence of parental vaccine hesitancy ranged from 20.1% to 26% in three studies conducted in Islamic countries [19,20,21]. In addition, 52.7% of parents in the United States reported a decline in parental willingness toward childhood vaccination after the COVID-19 outbreak compared to the time before COVID-19 (*p* < 0.01) [22].

Furthermore, the pooled prevalence of childhood/routine delayed vaccination was 38.2% (95% CI = 10.1–77.2%, four studies, *I*^2^ = 99%; Appendix A). In Switzerland, only 8% of the parents delayed the vaccination schedule of their children during the COVID-19 pandemic [23]. In three other studies, the prevalence of delayed childhood vaccination was 12.6%, 35.7%, and 38% during the COVID-19 pandemic, respectively [21,24,25]. However, a much higher proportion of parents reported delayed childhood vaccination in China (74.8%) [26].

Parents’ demographics, socioeconomic status, parental knowledge/attitude toward childhood vaccination, trust in information, and information source were factors associated with parental willingness of childhood vaccination in the included studies. Regarding parents’ demographics, older age (e.g., 45–54 years old, *p* = 0.012, [22]), higher education level (i.e., with a master’s degree, *p* = 0.039, [22]; *p* = 0.016, [27]; *p* = 0.026, [19]; *p* = 0.0288, [20]), higher household income [22], and being healthcare workers (*p* = 0.0001) [19] were associated with higher willingness for childhood/routine vaccination among parents. Being male was associated with higher vaccine hesitancy among Chinese parents (adjusted odds ratio [AOR] 1.372, *p* = 0.032) [16]. Ethnicity played a controversial role in childhood/routine vaccine hesitancy among parents. As compared to White parents, parents who were Hispanic or multiple races had lower childhood/routine vaccination hesitancy [22]. Parents having at least two children were more likely to delay childhood/routine vaccination [22]. Regional differences were also observed between Chinese cities (e.g., Shanghai vs. Wuhan [26]) and districts in Saudi Arabia [21]. Moreover, several studies revealed that better knowledge and positive attitude among parents were correlated with lower hesitancy/delay in childhood vaccination [20,24,27]. At the interpersonal level, lower confidence in the information provided by health staff [24] and being told by others that children had a bad reaction to the vaccine or vaccine was not safe [20] increased the likelihood of vaccine hesitancy or delaying vaccination for children. Sources to obtain information related to vaccination during the pandemic also influenced vaccine hesitancy among parents. A study in Saudi Arabia showed that obtaining information from a healthcare professional in person was associated with a higher delay in childhood vaccination [25], while a Turkish study suggested that obtaining information from health institutions and health workers decreased the likelihood of childhood vaccine hesitancy [19]. Obtaining information from family, friends, and spouses was associated with an increased likelihood of delaying childhood vaccination among parents [19,25]. The use of social media not only increased the level of fear among caregivers but also had a negative influence on their decisions about children’s vaccination [25]. Searches on social media (e.g., YouTube and Facebook) increased the odds of delaying vaccination [25]. Similar effects were also found for other written, audio, and visual media [19]. In Saudi Arabia, using the Ministry of Health (MOH)’s Sehha app or calling the MOH call centers to obtain relevant information decreased the odds of delaying childhood vaccination [25].

Furthermore, the main reasons for parents to accept childhood vaccination included (1) trust in the vaccination information given by health professionals (85.1% [24]); (2) belief that vaccines were important for children (95.4% [20]); and (3) perceived vaccines offered by the government were beneficial for children (94.6% [20]). The key reasons for childhood vaccination hesitancy in parents included (1) the belief that children received more vaccines than necessary (30.6% [24]; 28.3% [20]), and (2) concerns about side effects or safety issues of vaccines (57.7–63.6% [24]; 34.6–43.6% [19]; 44.3% [20]).

#### 3.2.2. Seasonal Influenza Vaccination

Studies consistently found a significant increase in the parental willingness of children’s seasonal influenza vaccination during the COVID-19 pandemic (Table 1). Based on the meta-analysis, the pooled prevalence of willingness towards seasonal influenza vaccination was 47.3% (95% CI = 25.3–70.5%, five studies, *I*^2^ = 99%; Appendix A). The residual analyses examined outliers in one study that reported the prevalence of willingness towards seasonal influenza vaccination [23]. Sensitivity analysis by removing the outlier shows an increase in the prevalence of willingness towards seasonal influenza vaccination (47.3% vs. 56.4%). One Chinese study found that 68.4% of parents were willing to have their children receive seasonal influenza vaccination [28], which was significantly higher than that in the time before COVID-19 (35.2%) [28]. Among these parents, 85.9% intended to have their children receive seasonal influenza vaccination if free vaccination was available [28]. Similar changes were observed in some other countries. In Switzerland, parents reported a more than 2-fold increase in the willingness of children’s seasonal influenza vaccination during the pandemic, as compared to the time before COVID-19 (17% versus 7.2%) [23]. About half (47.3%) of the parents in Saudi Arabia agreed to have their children receive seasonal influenza vaccination in the next seasons, which was higher than the time before COVID-19 (29.8%) [29]. In China, 54.7% of children received seasonal influenza vaccination in the season of 2019–2020, while 80.9% of parents intended to have their children receive such a vaccine in the next season [26]. A study conducted in six developed countries found that 54.2% of parents were willing to vaccinate their children against seasonal influenza, which was about 3-fold higher than the figure recorded before the time of COVID-19 (15.8%) [18].

Some sociodemographic characteristics of the children and/or parents were associated with parental willingness of children’s seasonal influenza vaccination during the COVID-19 pandemic. Having a daughter [29], younger age of the children [26], higher education level [18,26,29] and higher household/personal income among parents [29], parents being healthcare workers [29], and parents’ and/or children’s history of seasonal influenza vaccination [18,29,30] were associated with higher parental willingness. Poor health status [29] and the presence of medical conditions in children (e.g., asthma, atopy) [23,31] were also facilitators for parents to have their children receive seasonal influenza vaccination. Regional differences in parental willingness were also observed in Switzerland and Saudi Arabia [23,29]. Perceptions related to seasonal influenza vaccination also influenced parental willingness. Worry about children’s infection with influenza and trust in the medical system were associated with a higher parental willingness [29]. Variables related to COVID-19 also affected the parental willingness of seasonal influenza vaccination. COVID-19 infection among family members or neighborhoods [31] and perceived children were vulnerable to COVID-19 [18,29] were associated with higher parental willingness, while a negative association was found between children’s history of COVID-19 and the parental willingness [29].

Perceived low risk of infection (24.7%) was the most common reason for parents’ hesitancy for children’s seasonal influenza vaccination, followed by concerns about side effects (19.6%), and children’s fear of needles/syringes (19.4%) [29]. The main reasons for parents’ acceptance of pediatric influenza vaccination were doctors’ recommendations, belief in the protection of vaccines against seasonal influenza in children, and the COVID-19 pandemic [31].

#### 3.2.3. PCV

Meta-analysis was not performed towards PCV due to limited information. Two studies looked at PCV among children during the COVID-19 pandemic. In Turkey, 30.8% of parents had their children with asthma complete PCV between October 2020 and March 2021 [31]. However, only 12.8% agreed that PCV could protect children with asthma during the COVID-19 pandemic [31]. Over half (54.6%) of the parents in Mozambique indicated that their children had completed PCV [32]. Some qualitative reasons on whether to receive PCV were given by the parents. Adherence to the pediatric care services provided by health facilities was mentioned as a facilitator for children to receive PCV [32]. Barriers mentioned by the parents included: (1) social norms and limited family support placed the burden of childhood vaccination on mothers, (2) perceived poor quality of health services reduced parents’ trust in vaccination services, (3) concerns about side effects, and (4) parents hesitated to seed and advocate for vaccination due to power imbalances with health workers [32]. COVID-19 also created additional barriers for parents, such as social distancing, facemask requirements, supply chain challenges, and disrupted outreach services [32].

#### 3.2.4. HPV Vaccination

There was limited information in this session. Only one study investigated the psychological association and potential factors between willingness to accept HPV and COVID-19 vaccination in children and themselves [17]. The study revealed that parents’ perceived vulnerability of their children to HPV, perceived severity of HPV, and perceived response efficacy were positively and significantly associated with their acceptance of HPV or COVID-19 vaccination to their children.

### 3.3. Limitation of the Included Studies

Several limitations were reported by the included studies. Methodological issues included (1) being unable to establish a causal relationship due to the cross-sectional study design [21,23,30,33], (2) nonprobabilistic sampling that would limit the generalizability of the findings [16,17,18,20,21,22,25,26,27,28,29,30,31,32,33,34,35], (3) small sample size that would result in limited statistical power [22], (4) limited factors being investigated [16,25,28,35]. Self-reported data by parents/caregivers/guardians might cause recall bias, self-selection bias, and social desirability bias [16,18,22,26,27,28,29,31], and it might not represent children’s willingness [28,29]. Moreover, missing data [30] and a low response rate [23,26] also limited the transferability and usefulness of the studies. Finally, attrition of participants caused by the closure of the study venues during COVID-19 pandemics was a limitation [22,23,31,32]. Moreover, it is difficult for researchers to investigate the causal relationship between intention/uptake of vaccination and health authority policies [16,23,26,32].

## 4. Discussion

Although a recently published systematic review and meta-analysis estimated parents’ hesitancy to vaccinate their children against COVID-19 [36], there was a dearth of systematic review and/or meta-analysis looking at parents’ hesitancy for other childhood vaccination. As compared with the time before COVID-19, there was no evidence supporting significant changes in parental willingness of childhood/routine vaccination after the COVID-19 outbreak. However, studies consistently reported a significant increase in parental willingness/acceptance to vaccinate their children against seasonal influenza (1.48–3.43-fold higher) after the COVID-19 outbreak. Very low parental willingness for children’s PCV was found during the pandemic [31]. Acceptance of PCV for children was relatively low among parents before the time of COVID-19 [37,38]. Regarding parental willingness of children’s HPV vaccination, there were large variations across countries before the time of COVID-19 [39,40]. However, the limited number of studies was unclear about the impact of COVID-19 on children’s PCV and HPV vaccination. The decline in the number of administered doses during the pandemic might be due to the interruption in vaccination service provision (e.g., shortage of healthcare workers, shift in priority of medical services, and COVID-19 control measures such as lockdown).

Associated factors of the parental willingness of pediatric vaccination during the COVID-19 pandemic were categorized into three levels following the socioecological model [3]. Factors at the individual level included sociodemographic characteristics, vaccination history of both children and parents, perceived health status or vulnerability of children, and trust in healthcare providers/authorities. In many studies, higher education level was associated with higher parental willingness for children’s vaccination [19,20,22,27]. However, a mixed relationship between education level and parental willingness was found among Chinese parents. Those with better education were more likely to delay the children’s routine vaccination during the pandemic [16,26], but were more likely to vaccinate their children against seasonal influenza vaccination [26]. It is necessary for future studies to investigate the underlying reasons between parents’ education level and parental willingness. Moreover, insufficient awareness of children’s vaccination status may result in a low parental willingness [18,31]. Therefore, it is necessary to provide sufficient health information and reminders to parents in order to increase their awareness of children’s vaccination status. Factors at the interpersonal level included influences of others (relatives/friends/spouses) and traditional and social media. Similar to the findings of a study looking at information exposure and mental and behavioral health during the pandemic [41], different sources of information exposure had different effects on the parental willingness of children’s vaccination. Exposure to vaccination-related information through official channels was associated with a lower likelihood of delaying childhood vaccination. These media channels mainly reported information verified by expert sources. As compared to official channels, unofficial media sources contained not only factual information but also emotional contents [41]. Negative emotions are more likely to be amplified in these media during the COVID-19 outbreak. Information exposure through interpersonal communication also increased the likelihood of delaying routine vaccination. It is likely that fake and unverified information was disseminated through this channel [41]. Health authorities should make use of the official channel to disseminate information related to childhood vaccination during the pandemic. COVID-19-related factors also affected parental willingness during the pandemic. The perception of children as vulnerable to COVID-19 was associated with increased parental willingness to have their children receive other vaccines (e.g., seasonal influenza vaccination, PCV, and HPV vaccination). Developing synergetic strategies to promote COVID-19 vaccination together with other pediatric vaccination is warranted during the pandemic [17].

Currently, there is also a growing body of studies reporting the interventions to maintain and/or enhance the uptake of pediatric vaccination during the COVID-19 pandemic. Some approaches aimed to remove the barriers of parental decision-making on pediatric vaccination. One study emphasized continuous capacity-building for healthcare workers to enhance the quality of vaccination services and users’ trust in service providers [42]. Researchers highlighted that public health nurses should be proactive in advocating the policies and addressing parents’ concerns related to childhood vaccination during the COVID-19 pandemic [43]. The “Development of Systems and Education for Human Papillomavirus Vaccination” (DOSE), a capacity-building program for healthcare providers, was shown to be effective in improving HPV vaccination coverage during the COVID-19 pandemic [44]. Moreover, fear of COVID-19 infection and logistic barriers caused by COVID-19 and its control measures resulted in a decrease in children’s visits to healthcare facilities, which reduced vaccination opportunities for children and disrupted childhood vaccination implementation. Using online technologies to remind parents and update them about their children’s vaccination status may be a useful strategy during the COVID-19 pandemic [8,42,45]. A study conducted in India showed that a phone counseling intervention for parents increased vaccination uptake among children by more than 20% (from 65.2% to 88.2%) [46]. Furthermore, telemedicine assisted parents in making decisions and achieving self-efficacy by providing necessary and sufficient health information [8,42,43,45,46,47,48]. In Jordan, a mobile app called “Children Immunization App” helped child refugees and their parents to have health-risk and vaccination information in a challenging context (i.e., a refugee camp), which was indicated to be a feasible and innovative means during the COVID-19 pandemic [47]. A randomized controlled trial in Korea showed that parents who used a “Child Vaccination Chatbot Real-time Consultation Messenger Service” had higher score than the control group in terms of vaccination information, motivation, and self-efficacy in children’s vaccination and vaccination intention [48]. Finally, the removal of environmental barriers also made great improvements for children’s vaccination during the epidemic. A successful example was presented by four primary care clinics at Arkansas Children’s Hospital in Little Rock, USA [49]. It was documented that a drive-through pediatric vaccine clinic was established at multiple drive-through sites in the community (other than healthcare facilities) during the pandemic. Besides the reduction in risk of contraction with COVID-19 and the burden of healthcare facilities, a significantly shortened waiting time (5 ± 2.77 min) was recorded (*p* < 0.001) when it was compared with regular clinical practice (17 ± 9.57 min). In addition to the positive feedback from both children and their parents, an apparently increased trend in monthly uptake of children vaccination was also revealed (e.g., ~4000 in September 2018, ~4700 in September 2019, and 5033 in September 2022). Apart from this, providing free vaccination was potentially useful to increase parental willingness (e.g., pediatric influenza vaccination [28]). In line with these, the US CDC recommended that a robustness of communication between healthcare professionals and the target population would help to maintain and enhance the primary care service [50]; whereas the Canadian catch-up strategies of childhood vaccination also underlined that health providers and policy-makers should identify the missing cases across the life course, detect gaps in service delivery, develop a multipronged tailored strategy, and enhance communication, documenting, evaluation, and readjustment of the vaccination programs after the pandemic [51].

The review has some limitations. Limitations that existed in the pooled studies were inevitable. First, no causal relationship was established, as all included studies were cross-sectional. Second, self-reported data might reduce the validity and accuracy of the study conclusions. Third, articles were retrieved from eight electronic databases, and the selected studies were limited for reviewing because only those with specific terms mentioned in the title/abstract were screened for further analysis. Those in a non-English language, with the publications formed as a conference abstract, government report, textbook, or unpublished dissertation, were not included.

## 5. Conclusions

The current review summarized and analyzed the parental willingness of children’s vaccination (including childhood vaccination, seasonal influenza vaccination, HPV vaccination, and PCV) during the COVID-19 pandemic. Associated factors were also revealed and compared with the situation before the time of COVID-19. The existing interventions to maintain or enhance pediatric vaccination in the era of COVID-19 were also interpreted and discussed in this study. Based on the evidence provided by this review, it is necessary to design and implement the parental intervention programs of children’s vaccination promotion targeting particular local context/circumstance in different countries/regions. This may contribute to the protection of children from vaccine-preventable diseases during or after the COVID-19 pandemic.

## Figures and Tables

**Figure 1 vaccines-10-01453-f001:**
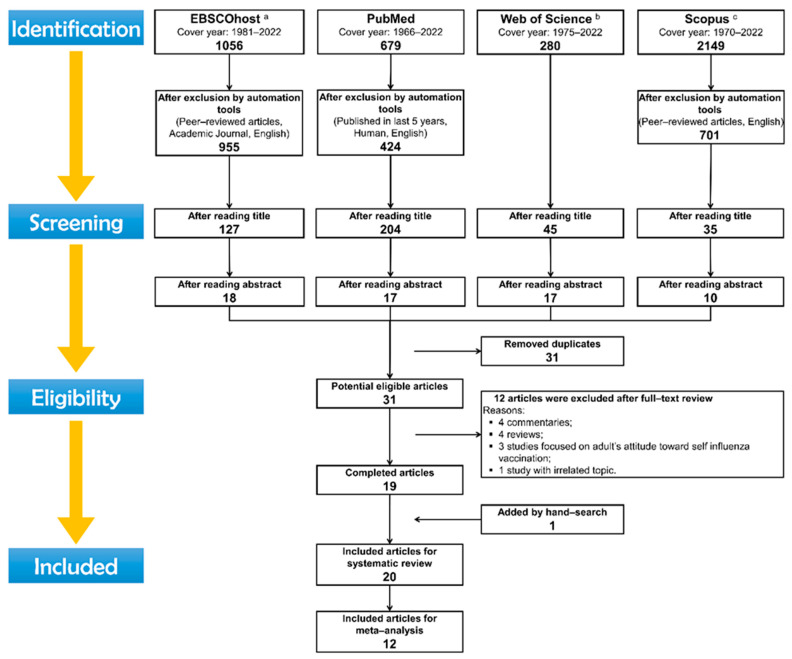
Preferred reporting items for systematic reviews and meta-analyses (PRISMA) flow diagram: search process for study selection about parental willingness to pediatric vaccination in the era of COVID-19. ^a^ Included databases of EBSCOhost are CINAHL with Full Text, ERIC, MEDLINE, APA PsycArticles, APA PsycInfo; ^b^ included databases of Web of Science are SSCI and A&HCI; ^c^ included both published papers and preprinted services.

**Figure 2 vaccines-10-01453-f002:**
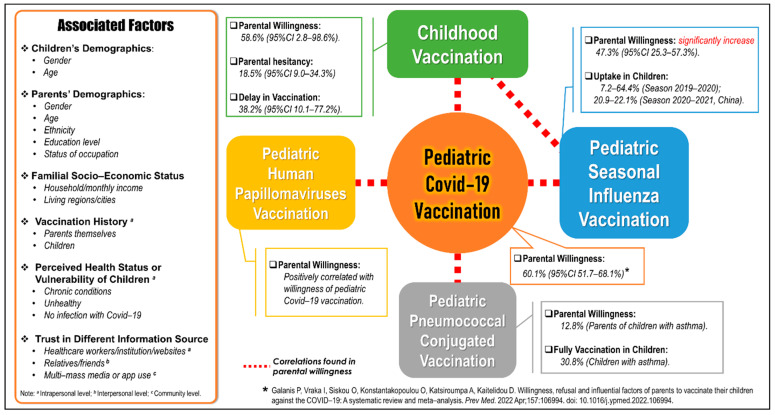
Parental willingness and associated factors to pediatric vaccination in the era of COVID-19.

**Table 1 vaccines-10-01453-t001:** Basic information of included studies.

No	Ref/Country	Study Design/Study Tool	Participants	Main Findings
Childhood/Routine Vaccination				
1	Wang et al., 2022 (China)	Three-wave repeated cross-sectional survey Vaccine Hesitancy Scale (VHS)	2881, 1038, and 1183 fathers or mothers with children aged ≤6 years were recruited from immunization clinics (aged 31.36 ± 4.38, 33.36 ± 4.74, and 32.12 ± 5.49 years, respectively; 69.5%, 89.1%, and 82.9% had an education level of college or above)	Prevalence of parental hesitancy to childhood/routine vaccination: 5.5–15.1%. Associated factors (for children): parents’ sex, time period of COVID-19 pandemic. Parental willingness to receive COVID-19 vaccination themselves: 59.3–92%. Associated factors (for self): parents’ sex, education level, healthcare occupation, number of contacts per day, perceived health status, and influenza vaccination history. Narrative reasons for parental vaccination hesitancy: concerns about vaccine safety and side effects.
2	Sinuraya et al., 2022 (Indonesia)	Cross-sectional survey A well-developed KAP questionnaire	276 parents (82% mothers, 53% aged > 38 years old, 54% being employed, 58% graduated from university, and 67.4% had children in kindergarten or primary school)	Positive correlations between parents’ knowledge, attitude, and practice for childhood/routine vaccination. Associated factors (for children): parents’ education level, occupation, knowledge, and attitude of childhood/routine vaccination.
3	He et al., 2022 (USA)	Cross-sectional survey Modified Vaccine Hesitancy Scale (VHS)	175 parents of children aged <18 years old (83.3% mothers, 55.2% aged 30–44 years old, 48.8% education level at high school, 52.4% household income < USD 49K, 61.9% Hispanic, 80.6% visited outpatient of general pediatrics)	Parental willingness to childhood/routine vaccination significantly decreased during COVID-19 compared with the situation before COVID-19 in terms of having a greater lack of confidence and perception of more risk. Associated factors (for children): parents’ age, education level, ethnicity, household income.
4	Gjini et al., 2022 (Albania)	Cross-sectional survey Self-developed questionnaire	475 parents of children aged 6 months to 8 years (89.6% mothers, 59.4% had education level at university or above, 85.1% aged 24 years or older, 34.3% had children at 0–6 months)	Prevalence of delay in childhood/routine vaccination: 12.6%. Parental willingness to adhere to vaccination schedule: 94.7%. Associated factors (for children): parental attitude toward childhood/routine vaccines, the risk of delay in a health center, trust in health staff; narrative reasons: parents play a role in children’s vaccination, belief of children receiving unnecessary vaccination, concerns about side effects of vaccines, trust in healthcare staff. Parental willingness to children’s COVID-19 vaccination: 23.8%.
5	Çağ et al., 2022 (Turkey)	Cross-sectional survey Self-developed questionnaire	1018 parents of children aged <13 years old (79.5% mothers, aged at 36.73 ± 6.67 years, mean number of children 2.05 ± 0.88, and mean age of the youngest child 5.52 ± 5.12 years)	Prevalence of parental vaccine hesitancy for children: 21.7% before the COVID-19 pandemic, 20.1% during the COVID-19 pandemic; associated factors (for children): parents’ education level, healthcare occupation, information source (i.e., healthcare professionals, multi-mass media, relatives/friends). Willingness to COVID-19 vaccination: 59% for parents themselves, 83% for children; associated factors (for children): parents’ vaccination history of seasonal influenza and COVID-19, parental willingness to receive COVID-19 vaccine for themselves.
6	Baghdadi et al., 2021 (Saudi Arabia)	Cross-sectional survey Self-developed questionnaire	577 caregivers of children aged ≤2 years (90.8% mothers, 93.6% Saudi nationals, 96.9% married women, mean-aged 32.6 ± 5.7 years old, 48.2% had private health insurance)	Prevalence of delay in childhood/routine vaccination due to COVID-19: 35.7%. Associated factors: use of social media, use of apps developed by healthcare authorities.
7	Al-Nafeesah et al., 2021 (Saudi Arabia)	Cross-sectional survey Self-developed questionnaire	1143 parents/guardians of children below six years (88% mothers/female; 75% mothers had bachelor’s degree, 20–39 years old, 39% had ≥3 children, 37% of parents resided in the Riyadh region)	Prevalence of delay in childhood/routine vaccination: 26% in regular situation, 38% during the COVID-19 pandemic. Associated factors: number of children in the family, living regions, parents’ reported delay in regular situations.
8	Aldakhil et al., 2021 (Saudi Arabia)	Cross-sectional survey SAGE working group Vaccination Hesitancy Scale	270 mothers (Aged 33 ± 5.5 years, 61.71% had completed bachelor’s degrees, 61.63% were employed, 85% had ≤4 children, 69.39% had children aged ≥18 months)	Prevalence of parental hesitancy to childhood/routine vaccination: 24.31%. Associated factors: parents’ education level, knowledge, attitude and awareness of vaccination, concerns about the safety of vaccines, trust in relatives/friends, trust in multi-mass media. Narrative reasons: concerns about the side effects of vaccines, adoption of the importance of vaccines for children’s health, financial consideration. Parental willingness to have their children vaccinated against COVID-19: 24%. Associated factors: parents’ education level.
**Influenza vaccination**				
9	Özer et al., 2022 (Turkey)	Cross-sectional survey Self-developed questionnaire	78 parents of children diagnosed with asthma (21.8% mothers and 28.2% fathers had education level at university or higher)	Uptake and intention rate of children’s seasonal influenza vaccine: 71.8% during COVID-19 (29.5% uptake before COVID-19, *p* = 0.001). Parental adoption of children’s uptake of seasonal influenza vaccine: 46.2%. Associated factors (for children): medical history related to asthma, contraction history of COVID-19 in family and close environment.
10	Hill et al., 2022 (USA)	Cross-sectional survey Self-developed questionnaire	299 parents of children aged <18 years old (87% mothers, aged 30–44 years old, 86% non-Hispanic White, 55% had household income > USD 99,999, 50% had children aged 6–12 years old)	Parents’ uptake rate of COVID-19 vaccine: 35%. Parental willingness to children’s COVID-19 vaccination: 46%. Associated factors (for children): parents’ self-vaccination history of COVID-19 and seasonal influenza, self-intention to COVID-19 vaccination. Parents’ self-vaccination history of seasonal influenza positively affected their willingness to vaccinate their children against seasonal influenza.
11	Du et al., 2022 (China)	Cross-sectional survey Self-developed questionnaire (with HBM items)	3011 reproductive women (including those who were mothers and those who had no pregnancy history, 1.35% lived in central China, 23.65% were 21–25 years old, and 54.83% had a bachelor’s degree)	Prevalence of parental influenza vaccine hesitancy for children: 9.13%. Associated factors (for children): perceived susceptibility, perceived barriers, perceived benefit.
12	Zhou et al., 2021 (China)	Cross-sectional survey Self-developed questionnaire	792 adult participants (including parents, mean age 33.3 ± 7.1 years, 77.5% females)	Willingness to receive seasonal influenza vaccines increased significantly following the COVID-19 pandemic among parents (57.8% vs. 8%, *p* < 0.001) and for their children (78.4% vs. 64.8%, *p* < 0.001)
13	Seiler et al., 2021 (Switzerland)	Cross-sectional survey Self-developed questionnaire	662 parents (no demographic details)	Majority of participating children (92%) adhered to the childhood/ routine vaccination schedule. Uptake of seasonal influenza in participating children in 2019–2020: 7.2%. Parental willingness to vaccinate their children against seasonal influenza: 17%. Associated factors (for children): children with chronic conditions, regional differences. Parental willingness to children’s COVID-19 vaccination: 59.2%. No regional difference.
14	Salawati et al., 2021 (Saudi Arabia)	Cross-sectional survey Self-developed questionnaire	2501 caregivers of children aged 6 months–18 years (58.0% mothers, 30.1% aged 36–45 years, 28.1% aged 26–35 years; 97.4% Saudi, 70.1% had a university degree, 31.4% reported monthly incomes of USD 1334–3200 and 24.4% reported monthly incomes of USD 3201–5333)	Parental willingness to children’s seasonal influenza vaccination: 29.8% before the COVID-19 pandemic, 47.3% during the COVID-19 pandemic. Associated factors (for children): parents’ education level, perceived health status of their children, perceived vulnerability of their children against COVID-19, trust in healthcare professionals, household income, and living regions. Narrative reasons: perceived risk of influenza infection in children, fear of needles/syringes. Parental uptake of COVID-19 vaccination: 83.4%. Parental willingness to children’s COVID-19 vaccination: 74.0%. Parents’ sources for COVID-19 vaccination include internet, health workers, and social media.
15	Hou et al., 2021 (China)	Cross-sectional survey Self-developed questionnaire	1655 parents with children aged 3 to 17 years (65.1% mothers, 74.0% had a bachelor’s degree or above, an average of 3.5 ± 1.30 members)	Prevalence of delay in childhood/routine vaccination: 74.8%. Associated factors: regional difference, parents’ education level. Children’s uptake of seasonal influenza vaccines: 54.7% (2019–2020). Parental willingness to children’s seasonal influenza vaccination: 80.9% in the future after the COVID-19 pandemic. Associated factors (for children): parents’ education level, children’s age.
16	Goldman et al., 2021 (US, Canada, Israel, Japan, Spain, Switzerland)	Cross-sectional survey Self-developed questionnaire	2422 caregivers accompanying their children aged 1–19 years old who arrived at the ED of 17 study centers (97% parents, caregivers’ mean age was 40.0 ±7.6 years and median age of the child was 8.3 ± 4.6 years)	Parental willingness to vaccinate their children against seasonal influenza: 15.8% before the COVID-19 pandemic, 54.2% during the COVID-19 pandemic. Associated factors (for children): children’s vaccination status, parents’ history of self-vaccination against seasonal influenza, perceived vulnerability of their children in emergency department.
17	Beatty and Villwock, 2021 (USA)	Cross-sectional survey Self-developed questionnaire	179 parents (88.3% mothers, 92.7% White, 79% a bachelor’s degree or higher, 83.8% had private insurance, 82.1% had an annual salary higher than USD 74,999)	51% of parents supported adding the influenza vaccine to the mandatory list for school reopening. Distrust was found significantly higher in those parents who objected to the addition and were concerned about the vaccine’s side effects. 96% of pro-addition parents and 24% of anti-addition parents reported willingness to have their children receive COVID-19 vaccination if the vaccines were proved as “safe and effective”.
18	Sokol and Grummon, 2020 (USA)	Cross-sectional survey Self-developed questionnaire	1893 parents and guardians of children aged 6 months to 5 years (69% mothers, 95% aged 18–44 years old, 57% college graduates, 34% White and 45% Black, 51% non-Hispanic, 52% middle or high income)	60% of parents changed their plans for children’s seasonal influenza vaccination due to the COVID-19 pandemic. Only 21% of parents reported that they were more likely to vaccinate their children due to the COVID-19 pandemic.
**Human papillomavirus (HPV) vaccination**				
19	Olagoke et al., 2022 (USA)	Cross-sectional survey Self-developed questionnaire	342 parents of adolescents who were unvaccinated for HPV (46% mothers, 87% White, 89% married, 75% education at college or more, 69% > USD 75,000 household income, 83% employed, 66% had a son)	Parental perceived vulnerability of their children to HPV, perceived severity of HPV, perceived response efficacy to HPV vaccine was positively related to parental willingness to vaccinate their children against HPV and/or COVID-19.
**Pneumococcal conjugate vaccination (PCV)**				
20	Powelson et al., 2022 (Mozambique)	In-depth interviews Qualitative data	10 mothers of children aged 25–34 months who were fully vaccinated and 22 mothers of children who were partially vaccinated (70–73% lived in rural area; aged 20–30 years old; 50–70% completed primary education); 12 health workers responsible for delivering immunizations	Uptake rate of DPT- HepB- Hib in participating caregivers’ children: 90.1% for dose 1, 68.1% for dose 2, 59.1% for dose 3. Uptake rate of PCV in participating caregivers’ children: 95.45% for dose 1, 77.3% for dose 2, 54.6% dose 3. Reasons for parental vaccine hesitancy: social norms and family support for mothers, trust in vaccination services, concern about side effects, and relationships between caregivers and health workers.
9	Özer et al., 2022 (Turkey)	Cross-sectional survey Self-developed questionnaire	78 parents of children diagnosed with asthma (21.8% mothers and 28.2% fathers had education level at university or higher)	The uptake rate of children’s PCV: 30.8%. Parental adoption of PCV protecting children with asthma during COVID-19: 12.8%. Narrative reason: the majority of parents lacked information or awareness of the PCV vaccine for their children.

## Data Availability

Data presented in this study are contained within the article and Appendix A.

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
