# Peer review of "Parental Willingness and Associated Factors of Pediatric Vaccination in the Era of COVID-19 Pandemic: A Systematic Review and Meta-Analysis"

_vaccines, 2022, doi:10.3390/vaccines10091453_

Round 1

Reviewer 1 Report

Authors perform a systematic review to investigate parental willingness and associated
factors of pediatric vaccination in the era of covid-19 pandemic.

The manuscript is well-written adding new information in a significant public health issue
such as children vaccination during the covid-19 pandemic. Please consider the following
comments in order to improve the quality of the manuscript.

Introduction

In line 55, please replace his with this.

Methods

Please, report the complete search strategy in a supplementary file instead of the
manuscript. In the manuscript, just report the key-words.

Do you consider including Scopus and pre-print services in your search strategy? Please,
clarify the reasons that you did not search in Scopus and pre-print services.

Please, describe more the tool that you used to assess the quality of studies.

Please remove inclusion and exclusion criteria from Figure 1.

I believe that a meta-analysis to estimate the overall prevalence of willingness and hesitancy
towards childhood/routine vaccination and seasonal influenza vaccination would add
valuable information in your systematic review.

Results

Regarding the way that you present the factors associated with parental
willingness/hesitancy of pediatric vaccination during 60 the COVID-19 pandemic there is
missing information. For example, readers cannot see if researchers in an article investigate
the relation between parental age and parental willingness/hesitancy of pediatric
vaccination but they did not find a statistically significant relation. Thus, dear Authors please
advise Table 3 in the following citation to present the information regarding the factors
associated with parental willingness/hesitancy of pediatric vaccination during 60 the COVID-
19 pandemic.

Galanis P, Vraka I, Siskou O, Konstantakopoulou O, Katsiroumpa A, Kaitelidou D. Willingness,
refusal and influential factors of parents to vaccinate their children against the COVID-19:

A systematic review and meta-analysis. Prev Med. 2022 Apr;157:106994. doi: 10.1016/j.ypmed.2022.106994.

I would like to see articles in Table 1 to be separated in the following categories for a better
understanding:

- childhood/routine vaccination

- seasonal influenza vaccination

- human papillomaviruses (HPV) vaccination, and

- pneumococcal conjugated vaccination (PCV)

Discussion

A similar systematic review has already published. Please consider this review especially in
the Discussion section. This review is the following:

Galanis P, Vraka I, Siskou O, Konstantakopoulou O, Katsiroumpa A, Kaitelidou D. Willingness,
refusal and influential factors of parents to vaccinate their children against the COVID-19: A
systematic review and meta-analysis. Prev Med. 2022 Apr;157:106994. doi:
10.1016/j.ypmed.2022.106994.

Please remove the question in the lines 284-285.

Please, try to avoid phrases like “heated academic discussion”, “dramatic increase”, etc.

Limitations section should be expanded. Please, consider the possible effect of bias on the
results, e.g. how could selection bias caused by non-probabilistic sampling could affect the
results

Reviewer 2 Report

The present work investigates the parental willingness and associated factors of pediatric vaccination (including seasonal influenza vaccination,  human papilloma viruses vaccination and pneumococcal conjugated vaccination)   during 14 the COVID-19 pandemic. 

I find this topic extremely interesting and believe the authors have addressed it competently.

I have only some minor comments: 

1. Is their systematic review registered ? 

2. The introduction of the article follows a clear structure.  However, a more accurate and detailed description of the reasons why the COVID 19 pandemic had a negative influence on pediatric vaccination is needed.

See for instance : https://www.who.int/news/item/15-07-2020-who-and-unicef-warn-of-a-decline-in-vaccinations-during-covid-19): « Even when services are offered, people are either unable to access them because of reluctance to leave home, transport interruptions, economic hardships, restrictions on movement, or fear of being exposed to people with COVID-19. Many health workers are also unavailable because of restrictions on travel or redeployment to COVID response duties as well as a lack of protective equipment. »

3. Why did the authors only focused on three vaccines (i.e. seasonal influenza vaccination, human papillomaviruses (HPV) vaccination, and pneumococcal conjugated vaccination (PCV) and excluded others which belong to childhood routine vaccination such as Polio or Chickenpox ?

4. Discussion 

This review suggests that «  As compared with the time before COVID-19, there was no evidence supporting significant changes in parental willingness of childhood/routine vaccination after the COVID-19 outbreak ». Consequently how the authors explain that « the COVID-19 pandemic disturbed vaccination implementation in children globally. »? This should be explicitly stated by the authors

Round 2

Reviewer 1 Report

The authors revised the manuscript according to the suggestions.